# Sound Radiation of Orthogonal Antisymmetric Composite Laminates Embedded with Pre-Strained SMA Wires in Thermal Environment

**DOI:** 10.3390/ma13173657

**Published:** 2020-08-19

**Authors:** Yizhe Huang, Zhifu Zhang, Chaopeng Li, Jiaxuan Wang, Zhuang Li, Kuanmin Mao

**Affiliations:** State Key Laboratory of Digital Manufacturing Equipment and Technology, Huazhong University of Science and Technology, Wuhan 430074, China; yizhehuang@hust.edu.cn (Y.H.); jeff.zfzhang@foxmail.com (Z.Z.); 15927673439@163.com (C.L.); wjx@hust.edu.cn (J.W.); lz_mse@hust.edu.cn (Z.L.)

**Keywords:** shape memory alloys, thermal environment, composite laminates, sound radiation

## Abstract

The interest of this article lies in the sound radiation of shape memory alloy (SMA) composite laminates. Different from the traditional method of avoiding resonance sound radiation of composite laminates by means of structural parameter design, this paper explores the potential of adjusting the modal peak of the resonant acoustic radiation by using material characteristics of shape memory alloys (SMA), and provides a new idea for avoiding resonance sound radiation of composite laminates. For composite laminates embedded with pre-strained SMA, an innovation of vibration-acoustic modeling of SMA composite laminates considering pre-stain of SMA and thermal expansion force of graphite-epoxy resin is proposed. Based on the classical thin plate theory and Hamilton principle, the structural dynamic governing equation and the frequency equation of the laminates subjected to thermal environment are derived. The vibration sound radiation of composite laminates is calculated with Rayleigh integral. Effects of ambient temperature, pre-strain, SMA volume fraction, substrate ratio, and geometrical parameters on the sound radiation were analyzed. New laws of SMA material and pre-strain characteristics on sound radiation of composite laminates under temperature environment are revealed, which have theoretical and engineering functional significance for vibration and sound radiation control of SMA composite laminates.

## 1. Introduction

Composite laminates as structural components have been extensively implemented in various fields of mechanical engineering such as aeronautics, astronautics, naval architecture, and automotive engineering. Composite plate offers advantages such as lightweight, high strength; but resulting in considerable vibration noise. Especially when the external excitation frequency is equivalent to or close to the natural frequency of the composite laminates, the strong structural resonance leads to more radiated noise. Shape memory alloy (SMA) reinforced composites are an extremely versatile class of materials which have the characteristics of larger internal forces, unique ability of changing its material properties, wide range of operational temperature, and high durability [1]. Though embedded with SMA wire to the composite material, the modal performance of composite laminates can be adjusted by the volume fraction and pre-strain of SMA without modifying the original shape dimensions and boundary conditions. This modification can effectively avoid the structure resonance and minimize the noise radiated into the surroundings.

Acoustic radiation of composite laminates is intimately related with the structural mode and vibration response. Over recent decades, the vibration of SMA composite plates has attracted the attention of some scholars. Rogers et al. [2] proposed utilizing SMA fibers modified structural performance of laminated plates via theoretically analyzed natural frequencies and mode shapes. Ostachowicz et al. [3] established a finite element model for predicting the first three natural frequencies subjected to several kinds of pre-strain. Malekzadeh et al. [4,5,6,7] developed the linear free vibration of hybrid laminated composite plates embedded with SMA fibers, employing the Ritz method, which lead to extracting the fundamental natural frequency, and investigated the dynamic behavior on detailed parametric analyses such as volume fraction, pre-strain, and aspect ratio. Park et al. [8] studied the vibration behavior of thermally buckled SMA composite plates using the von Karman nonlinear incremental strain–displacement relation. Furthermore, there are some investigations of the SMA composite plates with regard to vibration response, such as nonlinear vibration analysis [9,10,11], prediction of thermo-mechanical response under the combined action of thermal and mechanical loads [12,13], active vibration control of structures [14,15,16,17], and the experiment of structural acoustic characteristics [18].

Few studies about the acoustic radiation of SMA composite laminates are presented in the reported researching. However, research on vibration and acoustic radiation of conventional composite laminates has been very active. The design of material and structure parameters is an important means to improve the acoustic radiation. Comparing and analyzing sound radiation from material parameters of laminates, the conspicuousness of material composition on response, and the coupling effect of material proportion on sound radiation is often of great significance [19,20,21]. In terms of structural parameters, the research on sound radiation has developed from the design of normal geometrical parameters (aspect ratio, thickness) of laminates to more complex structural forms such as periodically reinforced structures and non-uniform laying [22,23,24,25]. In addition, as a current hotspot, the investigation of acoustic radiation of functionally graded laminates comprehensively considers the effects of materials, structures, external excitations, and the environment [26,27,28]. For this study, a new method to estimate sound power of orthogonal antisymmetric composite laminates embedded with SMA considering the pre-strain of SMA and thermal environment was presented and so the thermal expansion force and recovery force are added to the constitutive model of composite laminates. Based on the Hamilton principle and Rayleigh integration, the vibration differential equation and sound radiation calculation model of SMA composite laminates are established. The influence of SMA wires on the sound radiation power of laminates was examined by varying the pre-strain, SMA volume fraction, substrate ratio, and structural size parameters with temperature. The sound radiation results in the study provide a theoretical basis for vibration and sound radiation control of embedded SMA composite laminates.

## 2. Material Properties of SMA in Composite Laminates

Consider a rectangular composite laminates of length *L_a_*, width *L_b_*, and thickness *h* in a Cartesian coordinate system as shown in Figure 1.

Each layer of material is composed of SMA (nickel-titanium alloy), graphite, and epoxy in which SMA fibers are distributed in laminates in the form of orthogonal anti-symmetry. The elastic modulus of the SMA appears to have a strong temperature dependence [29]. From Figure 2, the modulus is 25 to 30 GPa when the temperature is below 40 °C as SMA in the martensitic phase.

As the temperature increases, the modulus of SMA increases rapidly when it transforms from the martensitic phase to austenite phase. Most of SMA are in the austenite phase when the temperature reaches 60 °C, and its modulus reaches about 80 GPa. After that, the modulus increases slowly with the rising temperature.

Recovery stress produced by SMA is not only temperature dependent, but also tensile pre-strain dependent. It can be seen from Figure 3 that the recovery stress increases with the increase of pre-strain after the temperature is higher than 60 °C [29].

Furthermore, the effect of pre-strain is not significant for normal or low temperature environment. To composite laminates embedded with SMA, owing to its temperature dependent material properties and pre-strain characteristics, the thermal environment should be fully considered in the process of its application. Especially through the control of material properties by adjusting the temperature, which has great potential for improving the vibration and sound radiation of the composite laminates.

## 3. Structural Dynamic Response

Assuming that the SMA orthogonal anti-symmetric composite laminate shown in Figure 1 contains *N* layers, the stress–strain relation of the *k*th layer can be given as [30], Equation (1):(1)$${\left\{\begin{array}{c} \sigma_{x}\\ \sigma_{y}\\ \tau_{xy} \end{array}\right\}}^{\left(k\right)}=\left[\begin{array}{ccc} {\tilde{Q}}_{11}^{\left(k\right)} & {\tilde{Q}}_{12}^{\left(k\right)} & 0\\ {\tilde{Q}}_{12}^{\left(k\right)} & {\tilde{Q}}_{22}^{\left(k\right)} & 0\\ 0 & 0 & {\tilde{Q}}_{66}^{\left(k\right)} \end{array}\right]{\left\{\begin{array}{c} \varepsilon_{x}\\ \varepsilon_{y}\\ \gamma_{xy} \end{array}\right\}}^{\left(k\right)}-\left[\begin{array}{ccc} {\bar{Q}}_{11}^{\left(k\right)} & {\bar{Q}}_{12}^{\left(k\right)} & 0\\ {\bar{Q}}_{12}^{\left(k\right)} & {\bar{Q}}_{22}^{\left(k\right)} & 0\\ 0 & 0 & {\bar{Q}}_{66}^{\left(k\right)} \end{array}\right]{\left\{\begin{array}{c} \alpha_{11}\\ \alpha_{22}\\ 0 \end{array}\right\}}^{\left(k\right)}\Delta T+{\left\{\begin{array}{c} \sigma_{x}^{r}\\ \sigma_{y}^{r}\\ 0 \end{array}\right\}}^{\left(k\right)},$$
where ${\tilde{Q}}^{\left(k\right)}$ is the transformed reduced stiffness of the *k*th layer of composite laminate embedded with SMA and ${\bar{Q}}^{\left(k\right)}$ is the transformed reduced stiffness of the *k*th layer of the substrate which is graphite-epoxy. The term $\sigma^{r}$ denotes the recovery stress in the SMA fiber direction. Other equations and terms in Equation (1) are completed in Appendix A. Utilizing the classic thin plate theory, the strains associated with the displacements are, Equation (2):(2)$$\left\{\begin{array}{c} \varepsilon_{x}\\ \varepsilon_{y}\\ \gamma_{xy} \end{array}\right\}=\left\{\begin{array}{c} \frac{\partial u_{0}}{\partial x}\\ \frac{\partial v_{0}}{\partial y}\\ \frac{\partial v_{0}}{\partial x}+\frac{\partial u_{0}}{\partial y} \end{array}\right\}-z\left\{\begin{array}{c} \frac{\partial^{2}w}{\partial x^{2}}\\ \frac{\partial^{2}w}{\partial y^{2}}\\ 2\frac{\partial^{2}w}{\partial x\partial y} \end{array}\right\},$$
where ‘$u_{0}$’, ‘$v_{0}$’ are the displacement components of the middle plane along the *x*, *y* coordinates, respectively and ‘w’ is the displacement components of along *z* coordinate.

Then, the strain energy $U_{1}$ is obtained as, Equation (3):(3)$$U_{1}=\frac{1}{2}\overset{N}{\underset{k=1}{\sum }}\int_{h_{k-1}}^{h_{k}}\underset{S}{\iint }\left\{\sigma_{x}\varepsilon_{x}+\sigma_{y}\varepsilon_{y}+\tau_{xy}\gamma_{xy}\right\}dxdydz.$$

The additional strain energy $U_{2}$ which consists of the strain energy caused by recovery stress and the thermal deformation energy can be expressed as, Equation (4):(4)$$U_{2}=\frac{1}{2}\underset{S}{\iint }\left\{\left(N_{x}^{r}-N_{x}^{T}\right){\left(\frac{\partial w}{\partial x}\right)}^{2}+2\left(N_{xy}^{r}-N_{xy}^{T}\right)\left(\frac{\partial w}{\partial x}\right)\left(\frac{\partial w}{\partial y}\right)+\left(N_{y}^{r}-N_{y}^{T}\right){\left(\frac{\partial w}{\partial y}\right)}^{2}\right\}dxdy,$$
where Equations (5) and (6):(5)$$\left\{\begin{array}{c} N_{x}^{T}\\ N_{y}^{T}\\ N_{xy}^{T} \end{array}\right\}=\overset{N}{\underset{k=1}{\sum }}\left[\begin{array}{ccc} {\bar{Q}}_{11}^{\left(k\right)} & {\bar{Q}}_{12}^{\left(k\right)} & 0\\ {\bar{Q}}_{12}^{\left(k\right)} & {\bar{Q}}_{22}^{\left(k\right)} & 0\\ 0 & 0 & {\bar{Q}}_{66}^{\left(k\right)} \end{array}\right]\left\{\begin{array}{c} \alpha_{11}\\ \alpha_{22}\\ 0 \end{array}\right\}\Delta Tdz,$$
(6)$$\left\{\begin{array}{c} N_{x}^{r}\\ N_{y}^{r}\\ N_{xy}^{r} \end{array}\right\}=\sum_{k=1}^{N}{\left\{\begin{array}{c} \sigma_{x}^{r}\\ \sigma_{y}^{r}\\ 0 \end{array}\right\}}^{\left(k\right)}dz.$$

The damping deformation energy $U_{3}$ can be given as, Equation (7):(7)$$U_{3}=\underset{S}{\iint }F_{d}w\left(x,y,t\right)dxdy,$$
in which the damping is $F_{d}=\lambda \dot{w}$ and λ is the viscous damping coefficient. The total potential energy of composite laminates can be written as, Equation (8):(8)$$U=U_{1}+U_{2}+U_{3}.$$

The kinetic energy of composite laminates can be calculated as, Equation (9):(9)$$T=\frac{1}{2}\overset{N}{\underset{k=1}{\sum }}\int_{h_{k-1}}^{h_{k}}\underset{S}{\iint }{\bar{\rho }}^{\left(k\right)}{\left(\frac{\partial w}{\partial t}\right)}^{2}dxdydz.$$

It is noteworthy in Equation (9) that the sound radiation of the composite laminates is mainly generated by bending vibration, so that the kinetic energy in the x and y directions can be neglected. For a transverse load q(x,y,t) on the surface of the laminates, the work it has done on the system can be obtained as, Equation (10):(10)$$W=\underset{S}{\iint }q\left(x,y,t\right)w\left(x,y,t\right)dxdy.$$

According to the Hamilton’s principle, the dynamic equations of the laminates can be derived by Equation (11):(11)$$\delta \Pi =\int_{t_{0}}^{t_{1}}\left(\delta T-\delta U+\delta W\right)dt=0.$$

Substituting Equations (8)–(10) into Equation (11), meanwhile considering the condition of orthogonal antisymmetric composite laminates, only $B_{11}$ and $B_{22}$ are not zero in $B_{ij}$, and $B_{22}=-B_{11}$. Then, the vibration equation of an orthogonal antisymmetric composite laminate embedded in SMA can be derived as, Equations (12)–(14):(12)$$A_{11}\frac{\partial^{2}u_{0}}{\partial x^{2}}+A_{66}\frac{\partial^{2}u_{0}}{\partial y^{2}}+\left(A_{12}+A_{66}\right)\frac{\partial^{2}v_{0}}{\partial x\partial y}-B_{11}\frac{\partial^{3}w}{\partial x^{3}}=0,$$
(13)$$\left(A_{12}+A_{66}\right)\frac{\partial^{2}u_{0}}{\partial x\partial y}+A_{66}\frac{\partial^{2}v_{0}}{\partial x^{2}}+A_{22}\frac{\partial^{2}v_{0}}{\partial y^{2}}+B_{11}\frac{\partial^{3}w}{\partial y^{3}}=0,$$
(14)$$\begin{array}{c} B_{11}\left(-\frac{\partial^{3}u_{0}}{\partial x^{3}}+\frac{\partial^{3}v_{0}}{\partial y^{3}}\right)+D_{11}\frac{\partial^{4}w}{\partial x^{4}}+2\left(D_{12}+2D_{66}\right)\frac{\partial^{4}w}{\partial x^{2}\partial y^{2}}+D_{22}\frac{\partial^{4}w}{\partial y^{4}}-\left(N_{x}^{r}-N_{x}^{T}\right)\frac{\partial^{2}w}{\partial x^{2}}\\ -\left(N_{y}^{r}-N_{y}^{T}\right)\frac{\partial^{2}w}{\partial y^{2}}+\lambda \frac{\partial w}{\partial t}+I\frac{\partial^{2}w}{\partial t^{2}}=q, \end{array}$$
where Equations (15)–(18):(15)$$A_{ij}=\overset{N}{\underset{k=1}{\sum }}{\tilde{Q}}_{ij}^{\left(k\right)}\left(h_{k}-h_{k-1}\right),$$
(16)$$B_{ij}=\frac{1}{2}\overset{N}{\underset{k=1}{\sum }}{\tilde{Q}}_{ij}^{\left(k\right)}\left(h_{k}^{2}-h_{k-1}^{2}\right),$$
(17)$$D_{ij}=\frac{1}{3}\overset{N}{\underset{k=1}{\sum }}{\tilde{Q}}_{ij}^{\left(k\right)}\left(h_{k}^{3}-h_{k-1}^{3}\right),$$
(18)$$I=\overset{N}{\underset{k=1}{\sum }}{\bar{\rho }}^{\left(k\right)}\left(h_{k}-h_{k-1}\right).$$

In this paper, mechanical boundary conditions of composite laminates are intended to be simply supported, Equation (19):(19)$$w=v_{0}=0\;at\;x=0,L_{a}\;w=u_{0}=0\;at\;y=0,L_{b}.$$

The state variables satisfying the simply supported boundary conditions of Equation (19) are assumed as Equation (20):(20)$$\{\begin{array}{c} u\left(x,y,t\right)=\overset{\infty }{\underset{m=1}{\sum }}\overset{\infty }{\underset{n=1}{\sum }}U_{mn}cos\frac{m\pi x}{L_{a}}sin\frac{n\pi y}{L_{b}}e^{j\omega t}\\ v\left(x,y,t\right)=\overset{\infty }{\underset{m=1}{\sum }}\overset{\infty }{\underset{n=1}{\sum }}V_{mn}sin\frac{m\pi x}{L_{a}}cos\frac{n\pi y}{L_{b}}e^{j\omega t}\\ w\left(x,y,t\right)=\overset{\infty }{\underset{m=1}{\sum }}\overset{\infty }{\underset{n=1}{\sum }}W_{mn}sin\frac{m\pi x}{L_{a}}sin\frac{n\pi y}{L_{b}}e^{j\omega t} \end{array},$$
where $U_{mn}$, $V_{mn}$, and $W_{mn}$ are the Fourier expansion coefficients of the solution, *m* and *n* are the half wave numbers in *x* and *y* directions, and ω is the circular frequency. Similarly, the load q(x,y,t) is assumed to be expanded into a double Fourier series form, Equation (21):(21)$$q\left(x,y,t\right)=\hat{q}\left(x,y\right)e^{j\omega t}=\overset{\infty }{\underset{m=1}{\sum }}\overset{\infty }{\underset{n=1}{\sum }}q_{mn}sin\frac{m\pi x}{L_{a}}sin\frac{n\pi y}{L_{b}}e^{j\omega t},$$
where $q_{mn}$ as the load coefficient can be given as, Equation (22):(22)$$q_{mn}=\frac{4}{L_{a}L_{b}}\left(\overset{\infty }{\underset{m=1}{\sum }}\overset{\infty }{\underset{n=1}{\sum }}\overset{L_{a}}{\underset{0}{\int }}\overset{L_{b}}{\underset{0}{\int }}\hat{q}\left(x,y\right)sin\frac{m\pi x}{L_{a}}sin\frac{n\pi y}{L_{b}}dxdy\right).$$

Substituting Equations (20) and (21) into Equations (12)–(14) and the vibration equation can be rewritten into a form as, Equation (23):(23)$$\left(\left[K\right]+j\omega \left[C\right]-\omega^{2}\left[M\right]\right)\left\{X_{mn}\right\}=\left\{F_{mn}\right\},$$
where $\left\{X_{mn}\right\}={\left\{U_{mn},V_{mn},W_{mn}\right\}}^{T}$ and $\left\{F_{mn}\right\}={\left\{0,0,q_{mn}\right\}}^{T}$ are displacement amplitude vector and load amplitude vector, respectively. Moreover, the mass matrix, damping matrix, and stiffness matrix is respectively expressed as, Equation (24):(24)$$\left[M\right]=\left[\begin{array}{ccc} 0 & & \\ & 0 & \\ & & I \end{array}\right],\;\left[C\right]=\left[\begin{array}{ccc} 0 & & \\ & 0 & \\ & & \eta \omega_{mn} \end{array}\right],\;\left[K\right]=\left[\begin{array}{ccc} K_{11} & K_{12} & K_{13}\\ K_{12} & K_{22} & K_{23}\\ K_{13} & K_{23} & K_{33} \end{array}\right],$$
in which the damping loss factor is, Equation (25):(25)$$\eta =\lambda /\left(\omega_{mn}\overset{N}{\underset{k=1}{\sum }}{\bar{\rho }}^{\left(k\right)}\right),$$
and $K_{ij}$ are given as, Equations (26)–(31):(26)$$K_{11}=-A_{11}{\left(\frac{m\pi }{L_{a}}\right)}^{2}-A_{66}{\left(\frac{n\pi }{L_{b}}\right)}^{2},$$
(27)$$K_{12}=-\left(A_{12}+A_{66}\right)\left(\frac{m\pi }{L_{a}}\right)\left(\frac{n\pi }{L_{b}}\right),$$
(28)$$K_{13}=B_{11}{\left(\frac{m\pi }{L_{a}}\right)}^{3},$$
(29)$$K_{22}=-\left[A_{66}{\left(\frac{m\pi }{L_{a}}\right)}^{2}+A_{11}{\left(\frac{n\pi }{L_{b}}\right)}^{2}\right],$$
(30)$$K_{23}=-B_{11}{\left(\frac{n\pi }{L_{b}}\right)}^{3},$$
(31)$$K_{33}=D_{11}\left[{\left(\frac{m\pi }{L_{a}}\right)}^{4}+{\left(\frac{n\pi }{L_{b}}\right)}^{4}\right]+2\left(D_{12}+2D_{66}\right){\left(\frac{m\pi }{L_{a}}\right)}^{2}{\left(\frac{n\pi }{L_{b}}\right)}^{2}+\left(N_{x}^{r}-N_{x}^{T}\right){\left(\frac{m\pi }{L_{a}}\right)}^{2}+\left(N_{y}^{r}-N_{y}^{T}\right){\left(\frac{n\pi }{L_{b}}\right)}^{2}.$$

Multiply both sides of the Equation (23) by the inverse matrix of the coefficient matrix, Equation (23):(32)$$\left\{X_{mn}\right\}={\left(\left[K\right]+j\omega \left[C\right]-\omega^{2}\left[M\right]\right)}^{-1}\left\{F_{mn}\right\}.$$

Solving Equation (32) obtains displacement amplitude in z direction of composite laminates, Equation (33):(33)$$W_{mn}=q_{mn}/\left\{\bar{\rho }\left[\left(\omega_{mn}^{2}-\omega^{2}\right)+j\eta \omega \omega_{mn}\right]\right\},$$
where $\omega_{mn}$ is the natural frequency which can be solved by the following frequency Equations (34):(34)$$det\left(\left[K\right]-\omega^{2}\left[M\right]\right)=0.$$

Substituting Equation (33) into the third equation of Equation (20), it can be obtained that the displacement components along z coordinate, Equation (35):(35)$$w\left(x,y,t\right)=\frac{\sum_{m=1}^{\infty }\sum_{n=1}^{\infty }q_{mn}}{\left\{\bar{\rho }\left[\left(\omega_{mn}^{2}-\omega^{2}\right)+j\eta \omega \omega_{mn}\right]\right\}}sin\frac{m\pi x}{L_{a}}sin\frac{n\pi y}{L_{b}}e^{j\omega t}.$$

The derivative of the displacement in Equation (35) with respect to time is utilized to obtain the amplitude of the (*m*, *n*) order modal vibration velocity of composite laminates, Equations (36):(36)$${\hat{U}}_{mn}\left(x,y\right)=\frac{j\omega q_{mn}}{\left\{\bar{\rho }\left[\left(\omega_{mn}^{2}-\omega^{2}\right)+j\eta \omega \omega_{mn}\right]\right\}}sin\frac{m\pi x}{L_{a}}sin\frac{n\pi y}{L_{b}}.$$

## 4. Sound Radiation Power

For rectangular composite laminates embedded with SMA, each unit in the laminates can be regarded as a source of external vibrational acoustic radiation. The radiation sound pressure of the laminates is obtained by radiating the sound pressure of each unit, and then performing Rayleigh integration. Assuming that the amplitude of the vibration velocity of the $ds^{\prime }$ in the laminates is ${\hat{U}}_{mn}\left(x^{\prime },y^{\prime }\right)$, the amplitude of the sound pressure at ds can be formulated as, Equation (37):(37)$$dP_{mn}\left(x,y\right)=j\frac{k\rho_{0}c_{0}}{2\pi r}{\hat{U}}_{mn}\left(x^{\prime },y^{\prime }\right)e^{-jkr}ds^{\prime },$$
where k,$\;\rho_{0}$, and $c_{0}$ are the sound wave number, the density of the air, and the sound velocity, respectively. r is the distance from unit $ds^{\prime }$ to unit ds. The sound pressure and sound intensity of all panel elements are radiated as being defined by [30], Equations (38) and (39):(38)$$P_{mn}\left(x,y\right)=j\frac{k\rho_{0}c_{0}}{2\pi }\underset{S}{\iint }\frac{e^{-jkr}}{r}{\hat{U}}_{mn}\left(x^{\prime },y^{\prime }\right)ds^{\prime },$$
(39)$${\hat{I}}_{mn}\left(x,y\right)=\frac{1}{2}R_{e}\left[P_{mn}\left(x,y\right){\hat{U}}_{mn}^{*}\left(x,y\right)\right],$$
where ${\hat{U}}_{mn}^{*}\left(x,y\right)$ is the conjugate of ${\hat{U}}_{mn}\left(x,y\right)$. Using Equation (39), the radiated sound power of the composite laminate can be formulated as [26], Equation (40):(40)$$\hat{W}=\frac{k\rho_{0}c_{0}}{4\pi }\overset{\infty }{\underset{m,n}{\sum }}\underset{S^{\prime }}{\iint }\underset{S}{\int }R_{e}\left[{\hat{U}}_{mn}\left(x^{\prime },y^{\prime }\right)\frac{e^{-jkr}}{r}\;{\hat{U}}_{mn}^{*}\left(x,y\right)\right]\;ds^{\prime }ds.$$

The radiated sound power is usually written in the form of sound power level in decibel, which is defined by, Equation (41):(41)$$L_{\hat{W}}=10\mathrm{lg}\left(\hat{W}/{\hat{W}}_{0}\right),$$
where ${\hat{W}}_{0}$ is the reference power and ${\hat{W}}_{0}=1\times 10^{-12}W$.

## 5. Results and Discussion

### 5.1. Materials and Verification

The analytical method is thus deployed to carry out several parametric studies to explore the acoustic response of composite laminates in thermal environments. The rectangular composite laminates are composed of ten layers, simply supported on all edges with dimensions of 0.4 m × 0.3 m × 0.008 m are considered as shown in Figure 1. Young’s modulus of elasticity and recovery stress for SMA were taken from Figure 2 and Figure 3, respectively. Poisson’s ratio, density, and thermal expansion coefficient of SMA are 0.3, 645 kg·m^3^, and 10.26 × 10^−6^ 1/°C, respectively. SMA is embedded in each layer with a volume fraction of 20%; graphite and epoxy are invoked as the substrate for a ratio of 1/9. The material properties of the substrate are given in Table 1.

In addition, the stacking sequence is [0°/90°/0°/90°/0°]_a_ and the damping loss factor of the laminates is set at 0.01. To verify the code of sound radiation characteristics of composite laminates embedded with SMA, the natural frequency and the sound radiation were calculated by the present method. The first example is the natural frequency from a rectangular laminated composite plate with SMA, which is taken from Park et al. [8]. The size of the plate is 0.38 m × 0.305 m × 0.002 m with simply supported boundary conditions for all edges. Table 2 shows the comparison of the first 3 natural frequencies of the laminated composite plate subjected to temperature at 25, 45, 65, 100, and 120 °C.

As can be seen from Table 2, the comparison certifies the correctness of natural frequencies that was calculated by the present method. The second example is sound radiation from antisymmetric cross-ply [0°/90°]_2_ laminated composite flat panel (0.5 m × 0.5 m × 0.02 m) subjected to unit harmonic point load at x = 0.125 m and y = 0.125 m which is taken from Sharma et al. [20]. Table 3 represents the material properties of the graphite-epoxy and glass-epoxy.

As shown in Figure 4, the calculated result has an excellent agreement with Sharma et al. [20]. Thus, the present analytical method was a sound choice for the study.

### 5.2. Investigating the Influence of Temperature Dependent Material Properties

Composite laminates embedded with SMA are temperature dependent. In order to depict the acoustic radiation power of the composite laminates in the thermal field, it assumes to be a uniform heating process. Figure 5 shows a three-dimensional diagram of the acoustic radiation power of composite laminates in the frequency range of 10 to 1000 Hz during the temperature rise from 15 to 155 °C.

As shown in Figure 5, it can be observed that there are four to five peaks of the sound power level corresponding to the natural frequencies. First and second modes shift to lower frequency and approach to each other with the heating case. Simultaneously, the frequency gap between the second mode and the third mode is widened. Moreover, the value of the sound radiation power level also varies and fewer peaks occur in the frequency range showed owing to the effect of heating. These significant effects stem from the temperature dependent material properties of the modulus and pre-strain of the SMA. To investigate further sound radiation characteristics of composite laminates embedded with SMA, representative temperatures such as 25 °C (SMA in martensitic phase) and 100 °C (SMA in austenite phase) were selected as conditions for subsequent study.

### 5.3. Investigating the Influence of SMA Pre-Strain

The influence of SMA pre-strain on the acoustic radiation of composite laminates is studied. Ensure that the substrate material ratio and SMA volume fraction are constant. The tensile pre-strains of SMA are treated as 1%, 3%, and 5%, respectively, to calculate the acoustic radiation power of composite laminates. 

Figure 6 and Figure 7 show the effects of pre-strain on the sound radiation of the composite laminates with SMA in 25 and 100 °C, respectively.

No considerable changes are seen in the sound power level under different pre-strain at 25 °C; however, the peaks of sound power level shift towards high frequencies with the increase of pre-strain at 100 °C. From Figure 2 and Figure 3, SMA in the austenite phase has higher modulus and recovery stress than that in the martensitic phase, which can explain the fact that extremely small recovery stress is generated by pre-strain at normal temperature since SMA is in the martensitic phase; but recovery stress increases gradually with rising temperature, which is due to the transformation of the martensitic phase to austenite phase. Moreover, it can be known from Equation (31) that the recovery stress has a positive correlation with the coefficient $K_{33}$ in the stiffness matrix. Therefore, the influence of recovery stress produced by tensile pre-strain of SMA at 25 °C can be ignored; nevertheless, at 100 °C, the increase of the stiffness of the composite laminates with the increase of recovery stress, results in the increase of the corresponding natural frequencies.

### 5.4. Investigating the Influence of SMA Volume Fraction

Materials volume fraction is an important component parameter of the plate materials for the acoustic properties. This section considers the effect of SMA volume fractions of 10%, 20%, 30%, and 40% on the sound radiation of composite laminates.

The variation of sound radiation power with different SMA volume fraction of the composite laminates is depicted in Figure 8 and Figure 9.

The plot Figure 8 reveals that at 25 °C, the modes shift towards low frequencies with an increase of SMA volume fraction. Modulus of SMA in the martensitic phase at ordinary temperature is less than that of substrate modulus as shown in Figure 2. The increase in SMA volume fraction leads to a decrease in the overall stiffness of the laminates, which bring about the decrease of natural frequencies. When the temperature is 100 °C, the modulus of SMA in austenite phase is greater than the substrate. As the SMA volume fraction increases, the stiffness of the laminates increases, so the first two modes shift towards high frequencies. Besides, third and fourth modes shift slightly towards low frequencies.

### 5.5. Investigating the Influence of Substrate Ratio

Considering the larger modulus of graphite in the substrate, the volume fraction ratio of graphite to epoxy studied in this section determines the modulus of the substrate. The volume fraction of SMA is 20%, so the volume fraction of substrate is 80%. A comparative study was carried out with the volume fraction ratio of graphite to epoxy (*V_f_*/*V_m_*) as 1/9, 2/8, 3/7, and 4/6.

The result in Figure 10 and Figure 11 illustrate influence of the substrate ratio on the sound radiation of composite laminates.

It is observed that the corresponding peaks of the sound power level shift towards the higher frequency domain when the volume fraction of graphite increases in Figure 10. The reason is that graphite has a larger modulus than that SMA in the martensitic phase which significantly affects the stiffness of the laminates at 25 °C. As can be seen from Figure 11, at 100 °C, increasing the volume fraction of graphite in the substrate has little effect on the first mode of the composite laminates, which is due to the influence of the modulus of SMA in the austenite phase on the stiffness increase. However, after the second mode, as the proportion of graphite increases, the peaks of sound power gradually shift towards the high frequency, i.e., the higher the volume fraction of graphite in the substrate, the higher the high-order modal frequency can be increased.

### 5.6. Investigating the Influence of Geometrical Properties

The geometric properties of composite laminates are studied from the thickness and aspect ratio. Figure 12 and Figure 13 display the variation of sound radiation power with constant length and width (*L_a_* = 0.4 m, *L_b_* = 0.3 m) but different thickness, that is, ℎ = 0.008 m, ℎ = 0.012 m, ℎ = 0.016 m, and ℎ = 0.02 m are considered.

From Figure 12 and Figure 13, the trend of mode shift caused by thickness variation is consistent for 25 and 100 °C. With the increase of thickness, the first and second mode shifts towards high frequencies; fewer peaks occur in the frequency range showed for thickness *h* > 0.012 m in both normal and high temperature, which is due to the fact that some higher-order modes have shifted above 1000 Hz. In addition, the increase of the thickness of composite laminates leads to a reduction of sound radiation, which is significantly below 400 Hz; however, the thickness has no significant effect on sound radiation in the high frequency region.

Similar to the plate thickness, aspect ratio is another structural geometric parameter of composite laminates. In order to study the effect of increasing aspect ratio on sound radiation, the length of the laminates is increased and the width is shortened while maintaining a constant area, i.e., 0.4 m × 0.3 m, 0.425 m × 0.283 m, 0.490 m × 0.254 m, and 0.6 m × 0.2 m. The results presented in Figure 14 and Figure 15 indicate that the aspect ratio has a significant effect on the acoustic radiation, in which the first mode shifts towards high frequencies with the increase of the aspect ratio.

The foremost reason is that the bending stiffness of composite laminates increases with the ratio of length to width under the condition of a certain area, which increases the modal frequency.

As mentioned above, the observations that the variation of thickness and aspect ratio affects sound radiation is determined by the variation of structural stiffness. However, temperature changes stiffness of laminates by affecting material properties, which is not a factor influencing the stiffness from structural geometric dimension. Therefore, the variation of structural geometric parameters has no remarkable law on acoustic radiation between normal and high temperature, but the trends are comparable.

## 6. Conclusions

Vibro-acoustic radiation of composite laminates which embeds SMA orthogonal antisymmetric has been developed. The constitutive equation, energy method, and Hamilton principle are utilized to derive the governing equations, and sound radiation is calculated by employing the Rayleigh integral approach. From the current investigation, the following conclusions are made:(1)The pre-strain and volume fraction of SMA in composite laminates have a significant impact on vibro-acoustic radiation. The higher the pre-strain of the SMA at high temperatures, the modes shift towards high frequencies. However, the volume fraction of SMA has opposite effects on the modalities of acoustic radiation between low and high temperatures. That is, as the volume fraction of SMA increases, modes shift towards high frequencies at elevated temperatures, and the opposite occurs at low temperatures.(2)Graphite in the base material has a larger modulus which affects the stiffness of the laminates. The effect of increasing volume fraction of graphite at low temperature is opposite to that of SMA on sound radiation. At this point, the modulus of graphite is larger than that of SMA, at which point the graphite content dominates the overall stiffness of the laminates.(3)Several geometric parameter studies on the effects of geometrical properties on sound radiation of composite laminates are carried out. It is noticed that composite laminates embedded SMA have similar characteristics of sound radiation with homogeneous laminates in the temperature range. Influence of geometric parameters on sound radiation is mainly dependent on structural stiffness rather than material properties, resulting in insensitivity to temperature.

The above conclusion fully demonstrates that the composite laminates embedded with SMA wires have excellent ability to adjust the mode frequency and change sound radiation characteristics. This study provides a basis for avoiding modal resonance and reducing vibration and sound radiation of composite laminates from the point of view of material design.

## Figures and Tables

**Figure 1 materials-13-03657-f001:**
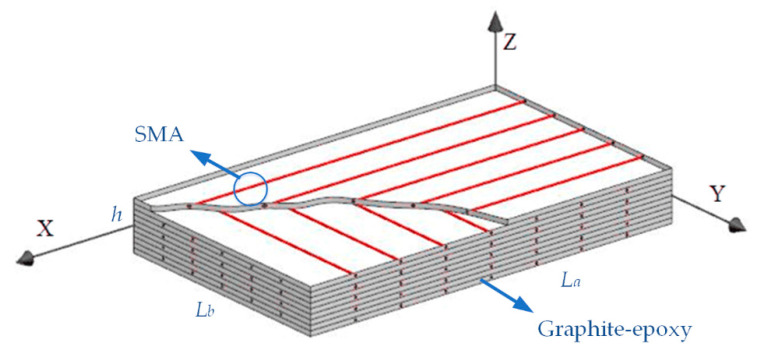
Schematic of orthogonal antisymmetric composite laminates embedded with shape memory alloy (SMA) wires.

**Figure 2 materials-13-03657-f002:**
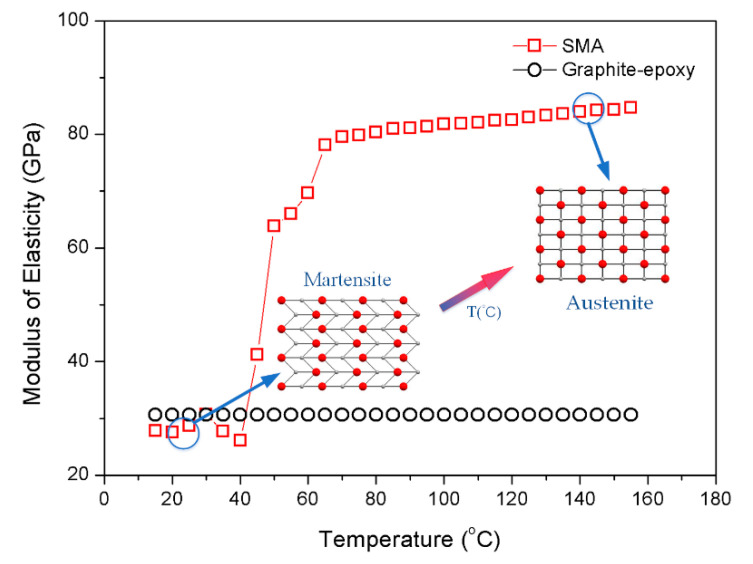
SMA modulus of elasticity vs. temperature.

**Figure 3 materials-13-03657-f003:**
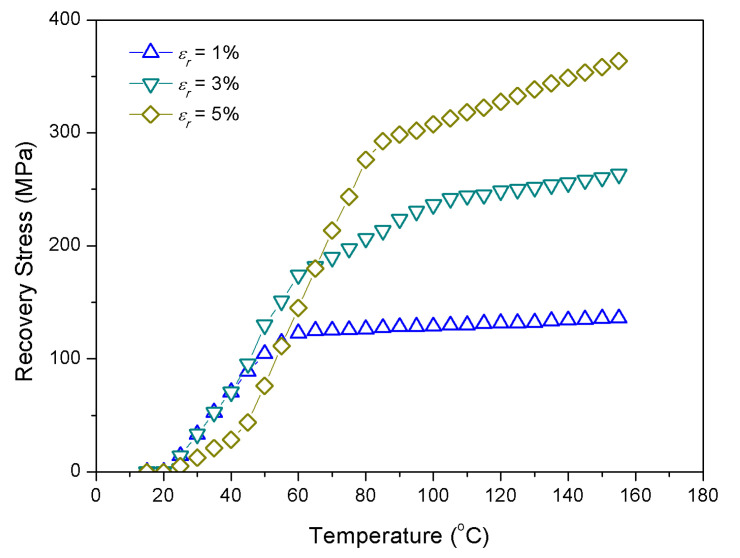
SMA recovery stress vs. temperature.

**Figure 4 materials-13-03657-f004:**
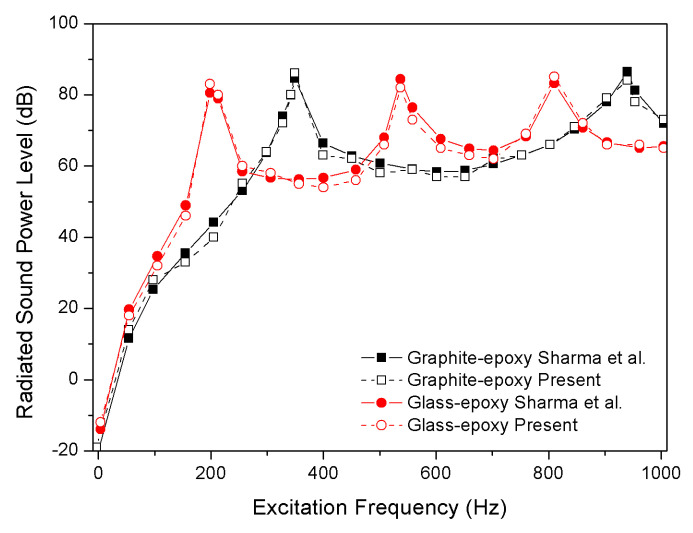
Comparison of sound power level with Sharma et al.

**Figure 5 materials-13-03657-f005:**
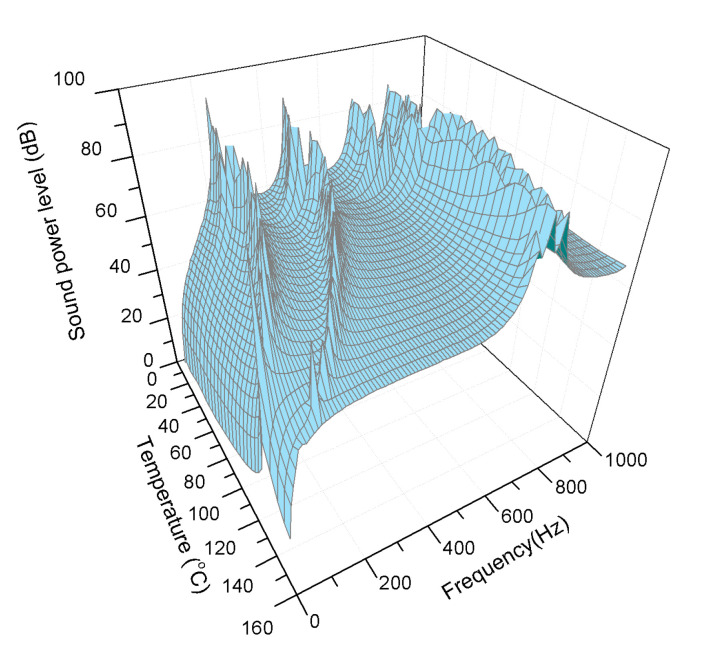
Sound radiation power of composite laminates embedded with SMA in the thermal environment.

**Figure 6 materials-13-03657-f006:**
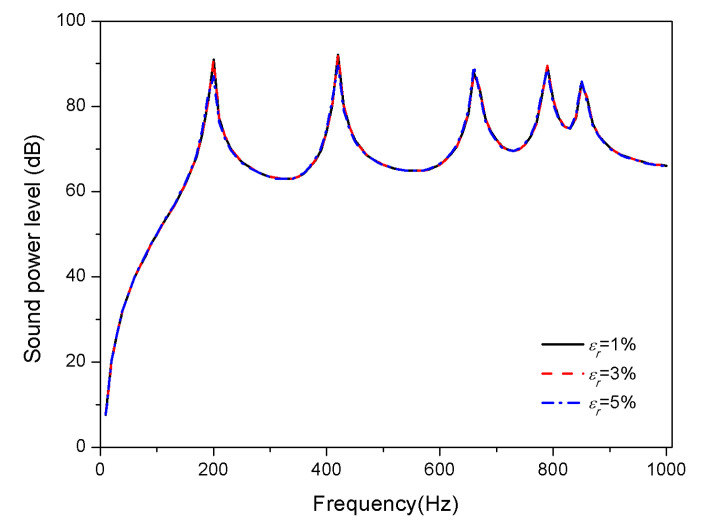
Sound radiation power of composite laminates embedded with SMA subjected to different pre-strain at 25 °C.

**Figure 7 materials-13-03657-f007:**
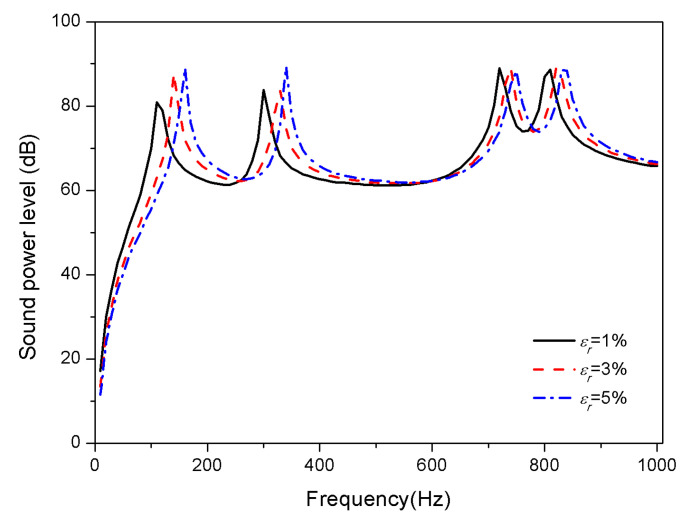
Sound radiation power of composite laminates embedded with SMA subjected to different pre-strain at 100 °C.

**Figure 8 materials-13-03657-f008:**
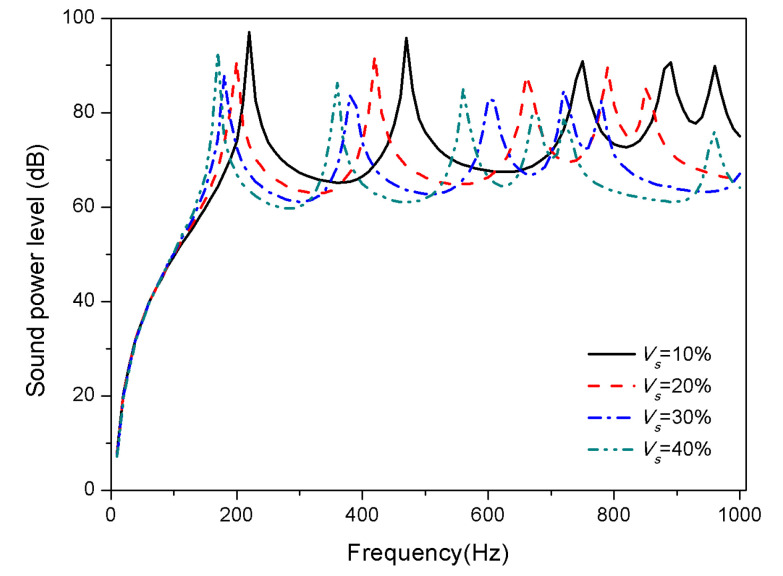
Sound radiation power of composite laminates embedded with SMA subjected to different SMA volume fraction at 25 °C.

**Figure 9 materials-13-03657-f009:**
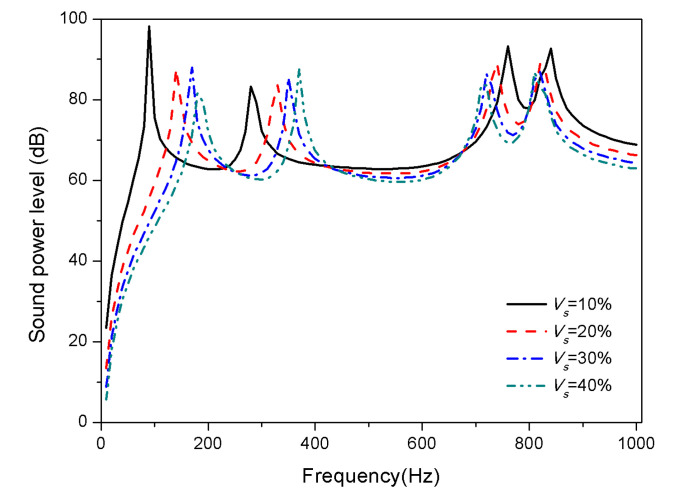
Sound radiation power of composite laminates embedded with SMA subjected to different SMA volume fraction at 100 °C.

**Figure 10 materials-13-03657-f010:**
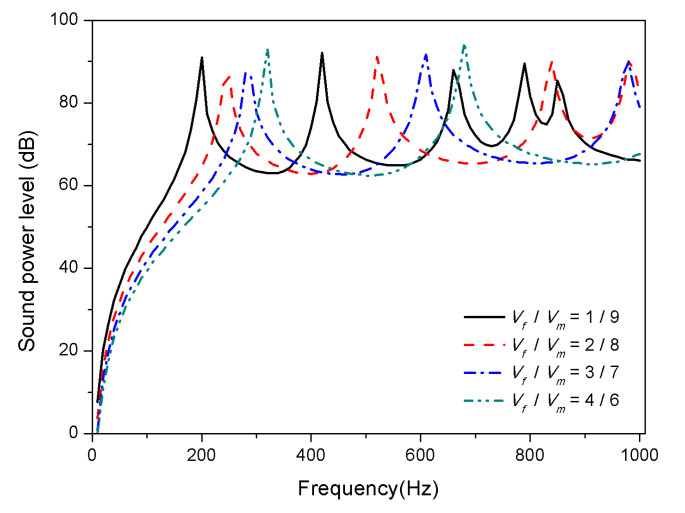
Sound radiation power of composite laminates embedded with SMA subjected to different substrate ratio at 25 °C.

**Figure 11 materials-13-03657-f011:**
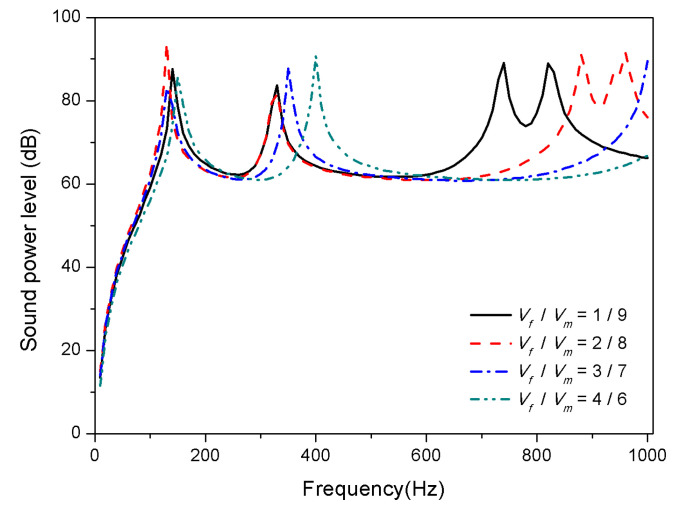
Sound radiation power of composite laminates embedded with SMA subjected to different substrate ratio at 100 °C.

**Figure 12 materials-13-03657-f012:**
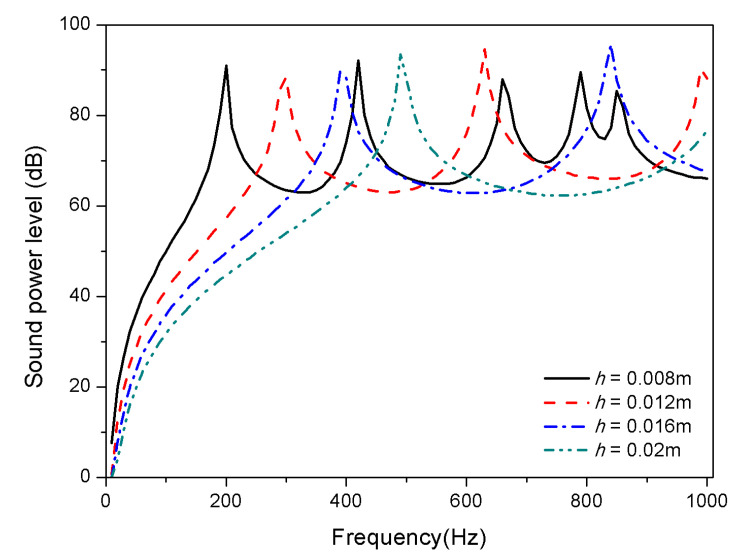
Sound radiation power of composite laminates embedded with SMA subjected to different thickness at 25 °C.

**Figure 13 materials-13-03657-f013:**
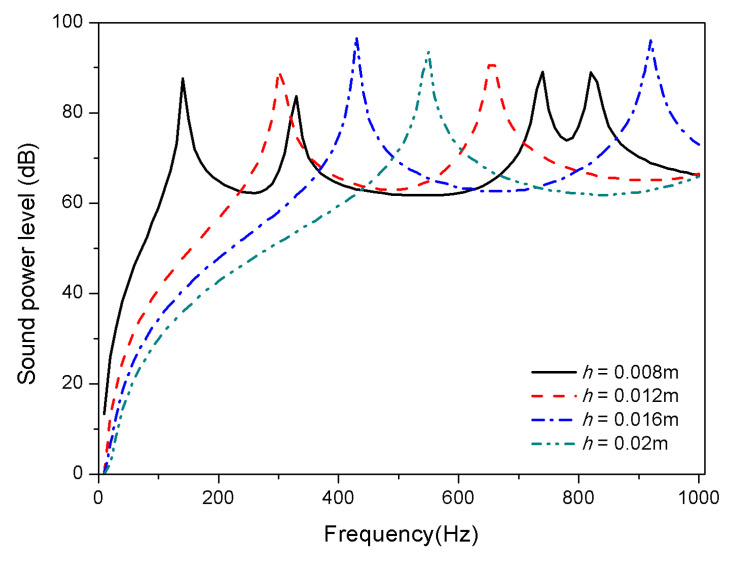
Sound radiation power of composite laminates embedded with SMA subjected to different thickness at 100 °C.

**Figure 14 materials-13-03657-f014:**
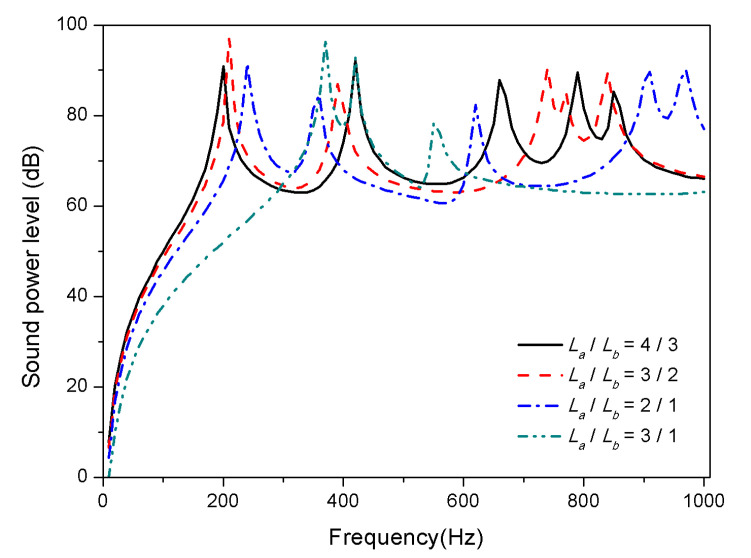
Sound radiation power of composite laminates embedded with SMA subjected to different aspect ratio at 25 °C.

**Figure 15 materials-13-03657-f015:**
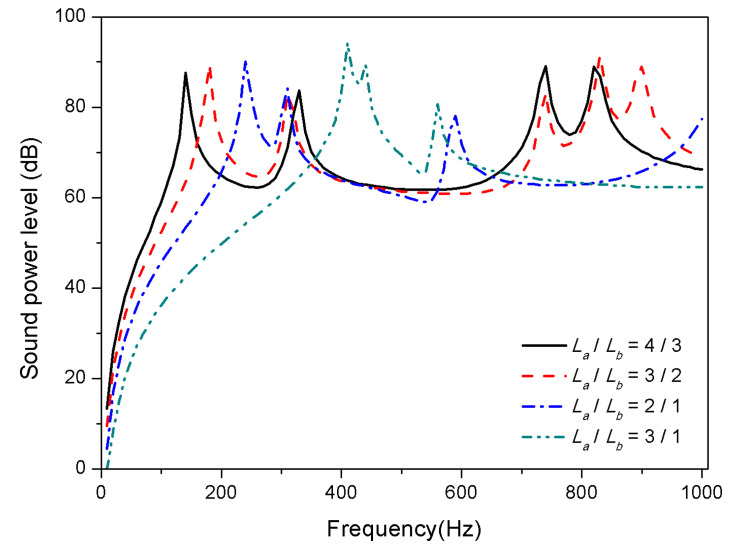
Sound radiation power of composite laminates embedded with SMA subjected to different aspect ratio at 100 °C.

**Table 1 materials-13-03657-t001:** Material properties of the substrate.

Substrate	$E_{1mf}\;(\mathrm{GPa})$	$E_{2mf}\;(\mathrm{GPa})$	$G_{mf}\;(\mathrm{GPa})$	$v_{12mf}$	$\alpha_{11}$ (1/°C)	$\alpha_{22}$ (1/°C)	$\rho_{mf}\;(\mathrm{kg}\cdot m^{3})$
graphite-epoxy	30.65	3.81	1.41	0.34	80.26 × 10^−6^	60.76 × 10^−6^	1315

**Table 2 materials-13-03657-t002:** Comparison of natural frequencies (Hz) with Park et al.

Modal Indices	Park et al.					Present				
	25 °C	45 °C	65 °C	100 °C	120 °C	25 °C	45 °C	65 °C	100 °C	120 °C
(1,1)	109.83	119.32	107.12	81.36	54.24	110.25	121.45	108.92	82.07	54.89
(2,1)	221.02	234.58	215.59	181.70	150.51	222.76	235.87	216.15	183.66	152.01
(1,2)	241.36	252.20	238.64	214.24	193.90	239.89	249.22	238.03	212.27	193.11

**Table 3 materials-13-03657-t003:** Composite material properties of graphite-epoxy and glass-epoxy.

Properties	Graphite-Epoxy	Glass-Epoxy
*E*_1_ (GPa)	137	38.6
*E*_2_ = *E*_3_ (GPa)	8.9	8.2
ν_12_ = ν_23_ = ν_13_	0.28	0.26
*G*_12_ = *G*_13_ (GPa)	7.1	4.2
*G*_23_ (GPa)	0.5*G*_12_	0.5*G*_12_
ρ (kg·m^3^)	1600	1900

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
