# Peer review of "Sound Radiation of Orthogonal Antisymmetric Composite Laminates Embedded with Pre-Strained SMA Wires in Thermal Environment"

_materials, 2020, doi:10.3390/ma13173657_

Round 1

Reviewer 1 Report

In the present manuscript, the authors have proposed a method to estimate sound power of orthogonal antisymmetric composite laminates embedded with shape memory alloy. I have serious concerns regarding the manuscript which are listed below:

  • It is not really clear what the authors wanted to achieve in the present manuscript. The idea of designing composites with shape memory alloys is not new. Several publications are already available on this topic. So, what is the novelty of the present study? Can the authors state clearly what their main achievements are?
  • From the equations presented in section 3, it is not really clear how the mutual interaction (i.e., elastic strain etc.) between the SMA and graphite-epoxy is addressed in the proposed approach. Can the authors describe it clearly? Also, how does elastic properties of SMA and graphite-epoxy, which should play important role in the formalism, enter into these equations? Can the authors comment on the role of defects such as dislocations etc.?
  • Which SMA was used for Fig2? Can the authors explain why the austenite phase has higher modulus of elasticity?
  • As Fig3 shows, recovery stress in the austenite phase varies significantly with varying pre-strain. In contrast, pre-strain has almost no effect on the recovery stress. Why?
  • What do the modes in Figs 6-15 physically mean? In Fig9, mode 1 and 2 show different dependence on SMA volume fraction than mode 3 and 4. What is the origin of this different behavior of modes?
  • In Fig 8 (martensite phase), the modes are equidistant while in Fig 9 (austenite phase), they are not. Different behavior of modes in the austenite and martensite phases is also observed in other cases, e.g., in Figs 10 and 11. What is the origin of this?

The current version of the manuscript has several other unclear issues. I am afraid that I cannot accept it.

Reviewer 2 Report

The paper presents an analytical solution for the sound radiation of composite panels with embedded SMA wires. The authors show that their solution nicely reproduces data from previous numerical studies available in literature. The model is then used to study the effect of several material and geometry parameters.

The work is interesting but it is difficult to understand the importance of the paper based on the abstract. I suggest the authours rewrite the abstract and clearly state that they derived an analytical model which compared well with previous numerical studies on the topic.

Further:

line 169: should refer to Fig.2 (?)

line 196, 199, 335: reference is missing

Reviewer 3 Report

The authors presented a study on composite plates embedded with shape memory alloys. They investigated the sound radiation levels using the classical plate modeling approach and Rayleigh integration. They did a thorough study of the effects of different parameters on the sound radiation. However, it is hard to identify the novelty in this study. The modeling approach is quite common, and the analysis is widely used in composite panels. They also mentioned that there are few studies on the acoustic radiation of SMA composite laminates. So, it is hard to identify the differences of those studies and the presented one. The authors should clearly identify the novelty in this study. 

Reviewer 4 Report

            It is correct scientific work about composite laminates embedded with pre-strained SMA wires in thermal environment and sound radiation. I recommend publication after minor revision.

Comments:

I recommend improving discussion from sections 5.2 to 5.6. There are no new references. It is necessary t0 compare results and general trend of the data with other studies from the literature of SMA laminates.

Why orthogonal antisymmetric option was selected? It can be justified with the optimization of composite response in thermal environment?

The peak positions of the sound power level versus frequency sometimes follows a linear trend?

Minor comments

The authors should check typography mistakes and omissions. As an example, the reference in line 196: with Sharma et al. [].

Figure 5: “...SMA in thermal.”?

Round 2

Reviewer 1 Report

The authors have carefully addressed all my concerns. I don't have any urgent queries for the authors. I believe that the modified manuscript can now be published.

Author Response

Thank you very much for your comments and affirmation of our manuscript.

Reviewer 3 Report

Based on the revision, I suggest the paper to be published. 

Author Response

(The authors gave the same response as above.)
